# Engineering *Escherichia coli* for Poly-β-hydroxybutyrate Production from Methanol

**DOI:** 10.3390/bioengineering10040415

**Published:** 2023-03-26

**Authors:** Jiaying Wang, Zhiqiang Chen, Xiaogui Deng, Qianqian Yuan, Hongwu Ma

**Affiliations:** 1Tianjin University of Science and Technology, Tianjin 300457, China; 2Biodesign Center, Key Laboratory of Engineering Biology for Low-Carbon Manufacturing, Tianjin Institute of Industrial Biotechnology, Chinese Academy of Sciences, Tianjin 300308, China; 3College of Food Science and Engineering, Tianjin University of Science and Technology, Tianjin 300457, China; 4School of Biological Engineering, Tianjin University of Science and Technology, Tianjin 300457, China; 5National Center of Technology Innovation for Synthetic Biology, Tianjin 300308, China

**Keywords:** methanol condensation cycle, non-oxidative glycolysis, polyhydroxybutyrate, methanol, *Escherichia coli*, methanol dehydrogenase gene

## Abstract

The naturally occurring one-carbon assimilation pathways for the production of acetyl-CoA and its derivatives often have low product yields because of carbon loss as CO_2_. We constructed a methanol assimilation pathway to produce poly-3-hydroxybutyrate (P3HB) using the MCC pathway, which included the ribulose monophosphate (RuMP) pathway for methanol assimilation and non-oxidative glycolysis (NOG) for acetyl-CoA (precursor for PHB synthesis) production. The theoretical product carbon yield of the new pathway is 100%, hence no carbon loss. We constructed this pathway in *E. coli* JM109 by introducing methanol dehydrogenase (Mdh), a fused Hps–phi (hexulose-6-phosphate synthase and 3-phospho-6-hexuloisomerase), phosphoketolase, and the genes for PHB synthesis. We also knocked out the *frm*A gene (encoding formaldehyde dehydrogenase) to prevent the dehydrogenation of formaldehyde to formate. Mdh is the primary rate-limiting enzyme in methanol uptake; thus, we compared the activities of three Mdhs in vitro and in vivo and then selected the one from *Bacillus methanolicus* MGA3 for further study. Experimental results indicate that, in agreement with the computational analysis results, the introduction of the NOG pathway is essential for improving PHB production (65% increase in PHB concentration, up to 6.19% of dry cell weight). We demonstrated that PHB can be produced from methanol via metabolic engineering, which provides the foundation for the future large-scale use of one-carbon compounds for biopolymer production.

## 1. Introduction

The increasing demand for energy and resources from rapid economic development has prompted research on possible alternatives. Some alternative sources, such as biogas, bioethanol, and biomethanol, have been widely produced from some waste materials [1,2]. Over 53 million tons of methanol, a potential, low-cost alternative substrate to sugar for bioindustries, is produced annually [3]. Studies on the use of methanol as a carbon source for the production of single-cell proteins, amino acids, and biopolymers have been intensively reported [4,5,6,7]. However, the commercial production of industrial chemicals from methanol using natural methylotrophs is limited mainly by the lack of well-developed genetic tools to implement synthetic pathways for objective products. Therefore, well-studied model strains, such as *Escherichia coli*, have gained great attention in the bioconversion of methanol [8]. There are three major natural pathways for methanol assimilation: the ribulose monophosphate (RuMP) cycle, the xylulose monophosphate (XuMP) cycle, and the serine cycle [9]. Among them, the RuMP cycle is considered the most applicable methylotrophic pathway in *E. coli*, as the expression of only three heterologous enzymes is sufficient for methanol assimilation [10]. Muller et al. introduced three RuMP cycle enzymes, including methanol dehydrogenase (Mdh), hexulose-6-phosphate synthase (Hps), and 6-phospho-3-hexuloisomerase (Phi), into *E. coli* to enable methanol assimilation [10]. Whitaker et al. engineered the RuMP pathway and the flavanone naringenin synthesis pathway in *E. coli*, demonstrating the possibility of in vivo conversion of methanol into a specialty chemical [8].

Although the RuMP cycle is the most applicable methanol assimilation pathway in *E. coli*, methanol utilization via the RuMP pathway involves CO_2_ loss and ATP expenditure in the synthesis of acetyl-CoA–derived products [11]. Specifically, three formaldehydes condense to pyruvate, which is decarboxylated to form acetyl-CoA and CO_2_, thereby reducing the maximum carbon yield of acetyl-CoA to 67%. Similarly, two other methanol assimilation pathways—the serine pathway and the XuMP pathway—also reduce the carbon yield to 54% and 67%, respectively, which are not optimal pathways for maximizing the yields of acetyl-CoA–derived metabolites [12]. To solve this carbon-loss problem, Bogorad et al. [11] designed a new methanol assimilation pathway, referred to as the methanol condensation cycle (MCC), which consists of RuMP with a non-oxidative glycolysis (NOG) pathway and can produce one molecule of acetyl-CoA from two molecules of methanol without carbon loss. This new pathway allows the efficient conversion of methanol to multi-carbon acetyl-CoA–derived chemicals with 100% theoretical carbon yield.

As a derivative product of acetyl-CoA, poly-3-hydroxybutyrate (PHB) has excellent biodegradability. Within a certain period of time, microorganisms can break down PHB into CO_2_ and H_2_O, resulting in a significant reduction in “white pollution” [13,14]. PHB also has very good biocompatibility and is mainly used in drug delivery carriers and biological tissue engineering [15,16,17]. Additionally, recent studies have indicated that not only is PHB crucial for carbon storage, but it also plays an important role in supporting bacterial survival under adverse conditions, such as intermediate metabolite toxicity [18]. To this end, in this study, we aim to engineer the MCC pathway in *E. coli* and convert methanol to the acetyl-CoA–derived product PHB. An extended metabolic network model of *E. coli* incorporating the methanol assimilation pathway and the PHB synthesis pathway is used to identify the optimal pathway. Then, the functional modules for methanol assimilation, PHB synthesis, and NOG are introduced into *E. coli*, thereby constructing the entire bioconversion pathway from methanol to PHB.

## 2. Materials and Methods

### 2.1. Metabolic Network Analysis

An extended genome-scale *E. coli* metabolic network model based on iML1515 was used to calculate the optimal production pathway and yield of PHB by performing flux balance analysis (FBA) [19,20]. PHB is a polymer; thus, the monomer 3-hydroxybutyrate of PHB was considered the objective product in FBA. As no methanol utilization and MCC pathways exist in *E. coli*, the methanol utilization reaction converting methanol to formaldehyde and reactions of the MCC pathway (*Mdh*, *Hps-phi*, and *Fxpk* (phosphoketolase)) were added to the iML1515 model. The FBA calculation was performed using the COBRApy [21]. In the calculation, the uptake rate of methanol was set to 6 mmol/gDCW/h. The non-growth-associated energy maintenance parameter was set to 0 mmol/gDCW/h.

### 2.2. Plasmid Construction

The primers used in the plasmid construction were obtained from GENEWIZ (Suzhou, China) and are listed in Appendix A. The genes *mdh2* (GenBank: CP007739.1, protein ID: AIE59127.1) from *Bacillus methanolicus* MGA3, *mdh-CT* (GenBank: CP002878.1, protein ID: AEI80320.1) from *Cupriavidus necator* N-1 [22], and *TaADH* (GenBank: AL445067.1, protein ID: CAC12437.1) from *Thermoplasma acidophilum* DSM 1728 [23] were synthesized by Qinglan Biotech (Wuxi, China) with codon optimization and then cloned with primers 01, 02, 03, 04, 05, and 06 with restriction sites (Table 1, Column 3) at the end. Then cloned genes were introduced into the pET-28a vector by enzyme-cut and linked up with corresponding restriction enzymes and ligases (TaKaRa, Kusatsu, Japan). The recombinant plasmids (Table 1) were confirmed by restriction mapping and sequencing (GENEWIZ, Suzhou, China). The confirmed constructs were subsequently transformed into *E. coli* BL21 (DE3) for protein expression. The fused Hps–phi gene was synthesized (Qinglan Biotech, Wuxi, China) according to the sequence reported by Orita et al. [24]. PCR experiments were performed using GoTaq DNA polymerase (TaKaRa, Kusatsu, Japan) with corresponding primers. Positive transformants were verified by colony PCR and Sanger sequencing.

### 2.3. Reconstruction of Expression Vector

Two *T7* promotors in the expression vector pACYCDuet-1 were replaced by the *Trc* promotor from pTrc99a to reconstruct a new vector called pZQ for the C1 assimilation module according to Golden Gate cloning [25]. The vector pACYCDuet-1 was first lined by amplification from the position of the *T7* promotor of the *mcs1* region with primers 09 and 10 (Appendix A), which were designed to exclude the sequence of the *T7* promotor. The lined sequence re-amplified by primers 11 and 12 (Appendix A) was designed to contain the half sequence of the *Trc* promotor and the restriction site of the type IIs endonuclease *BsaI* (Thermo Fisher Scientific, Waltham, MA, USA). The re-amplified sequence was then re-circled by enzyme-cut and link-up with *BsaI* and T4 DNA-ligase (Thermo Fisher Scientific, Waltham, MA, USA). Finally, the re-circled plasmid was finally transformed to *E. coli* JM109, and positive vectors were verified by colony PCR and Sanger sequencing.

Similarly, the second *T7* promotor was operated based on the first *T7* promotor. The differences between the two were that (i) the initial vector was pACYCDuet-1, in which the *T7* promotor changed to the *Trc* promotor, and (ii) the primers for the lining vector and re-amplifying primers (13, 14, 15, and 16; Appendix A) were designed to be longer to avoid amplifying the *mcs1* region.

### 2.4. Gene Modules Construction

A C1 assimilation module was built by introducing methanol dehydrogenase (Mdh), hexulose-6-phosphate synthase (Hps), and 3-phospho-6-hexuloisomerase (Phi) into a reconstructed plasmid pZQ. Each methanol dehydrogenase gene (*mdh2*, *mdh-CT*, and *TaADH*) was first introduced into the multiple cloning site 1 of plasmid pZQ, individually with primers 17 and 18, 19 and 20, and 21 and 22, respectively (the restriction enzyme cutting sites are listed in Table 1 and Table 2). The positive vectors were verified by colony PCR and Sanger sequencing. The multiple cloning site 2 of the vector pZQ with methanol dehydrogenase (Table 1) was added by Phs–Phi fusion with primers 23 and 24 (Appendix A), and the positive vector was verified by colony PCR and Sanger sequencing, after which the C1 assimilation module was constructed.

The NOG module was constructed on plasmid PBHR68 containing the PHB production module [26]. The fxpk gene from Bifidobacterium adolescentis was introduced into PBHR68 by using primers 25 and 26 (Table 1) with enzyme cutting and combination.

**Table 1 bioengineering-10-00415-t001:** Plasmids used in this study. The first and second columns contain the plasmids and genes used in this study, respectively. The third column contains the plasmid information.

Plasmid	Encoded Gene	Description	Reference
pET-28a		High-copy-number, *T7* promotor, pBR322 ori, Kan^R^	[27]
pET-28a_mdh2	*mdh2* gene from *B. methanolicus* MGA3	The *mdh2* gene was amplified from synthesized *mdh2* with codon optimization using primers 01 and 02 and ligated into the *NheI*/*BamHI* site of pET-28a.	[27]
pET-28a_mdh-CT	*mdh-CT* gene from *Cupriavidus necator* N-1	The *mdh-CT* gene was amplified from synthesized *mdh-CT* with codon optimization using primers 03 and 04 and ligated into the *NheI*/*HindIII* site of pET-28a.	This study
pET-28a_TaADH	*TaADH* gene from *Thermoplasma acidophilum* DSM 1728	The *TaADH* gene was amplified from *Thermoplasma acidophilum* DSM 1728 using primers 05 and 06 and ligated into the *NheI*/*HindIII* site of pET-28a.	This study
pET-28a_hps-phi	The artificial fusion of *hps* and *phi* with codon optimization	The artificial fusion of the *hps* and *phi* genes was amplified using primers 07 and 08 and ligated into the *NdeI*/*XhoI* site of pET-28a.	This study
pACYCDuet-1		High-copy-number, *T7* promotor, P15A ori, Cm^R^	Novagen
pTrc99a		High-copy-number, *Trc* promotor, pBR322 ori, Amp^R^	Novagen
pZQ		The two *T7* promotors of pACYCDuet-1 were replaced by the *Trc* promotor of pTrc99a using primers 09, 10, 11, 12, 13, 14, 15, and 16 according to the method of GGA.	This study
pZQ_mdh2	*mdh2* gene from *B. methanolicus* MGA3	The *mdh2* gene was amplified from synthesized *mdh2* with codon optimization using primers 17 and 18 and ligated into the *BamHI*/*EcoRI* site of pZQ.	This study
pZQ_mdh2-hps-phi	*mdh2* gene from *B. methanolicus* MGA3 and artificial fusion of *hps* and *phi* with codon optimization	The artificial fusion of the *hps* and *phi* genes was amplified using primers 19 and 20 and ligated into the *NdeI*/*XhoI* site of pZQ_mdh2.	This study
pBHR68	*phaA, phaB,* and *phaC* from *Ralstonia eutropha*	*phaA, phaB,* and *phaC* expression plasmid, pBluescript SK¯ derivative, Amp^R^	[26]
pBHR70	*phaA, phaB, phaC, fbp* and *fxpk* from *Bifidobacterium adolescentis*	*phaA, phaB, phaC, fbp* and *fxpk* expression plasmid, pBluescript SK¯ derivative, Amp^R^	This study
pTKRED		pSC101 replication, *ParaBAD*-driven I-*SceI* gene, λ-Red, Sp^R^	[28]
pTKS/CS		p15A replication, LP regions, *I-SceI* restriction sites, Cm^R^, Tet^R^	[28]

### 2.5. Genome Manipulation

The wild-type JM109 and the previously constructed *E. coli* strain JM109-NOG were used as the chassis strain for further genome modification [29]. Plasmids pTKS/CS and pTKRED were used to delete the gene *frmA* (glutathione (GSH)-dependent formaldehyde dehydrogenase); this deletion method was conducted according to a 1-step inactivation of chromosomal genes [30]. The functions and description of plasmids (pTKS/CS and pTKRED) used for genome manipulation are listed in Table 1. Corresponding primers (27, 28, 29, 30, 31, and 32) are listed in Appendix A. The strains used in this study are shown in Table 2.

**Table 2 bioengineering-10-00415-t002:** Strains used in this study. The first column reports the strains constructed in this study, the second column demonstrates the characteristics of the strains, and the third column shows the sources of these strains.

Strain	Feature	Source
JM109	*E. coli*, *recA1*, *endA1*, *gyrA96*, *thi*, *hsdR17*, *supE44*, relA1, Δ(*lac proAB*)/*F’* [*traD36*, *proAB^+^*, *lac^q^ lacZ*ΔM15]	TransGen Biotech
BL21(DE3)	*E. coli*, F–, *omp*T, *gal*, *dcm*, *lon*, *hsd*SB(rB- mB-), λ(DE3 [*lac*I *lac*UV5-T7 *gene* 1 *ind*1 *sam*7 *nin*5])	TransGen Biotech
BL21-pet28a_mdh2	BL21(DE3), pet28a_mdh2	This study
BL21-pet28a_mdh-CT	BL21(DE3), pet28a_mdh-CT	This study
BL21-pet28a_TaADH	BL21(DE3), pet28a_TaADH	This study
BL21-pet28a_ hps-phi	BL21(DE3), pet28a_ hps-phi	This study
JM109-Δfrm	*E. coli* JM109, Δ*frmA*	This study
JM109-Δfrm-mdh2	JM109-Δfrm, pZQ_mdh2	This study
JM109-Δfrm-C1	JM109-Δfrm, pZQ_mdh2-hps-phi	This study
JM109-NOG	JM109, P*_fbp_*::P*_J23100_*, ::*fxpk*	[29]
JM109-NOG-Δfrm	JM109-NOG, Δ*frmA*	This study
JM109-NOG-Δfrm-PHB	JM109-NOG-Δfrm, pBHR68	This study
JM109-NOG-Δfrm-C1-PHB	JM109-NOG-Δfrm, pZQ_mdh2-hps-phi and pBHR68	This study
JM109-Δfrm-C1-PHB	JM109-Δfrm, pZQ_mdh2-hps-phi and pBHR68	This study
JM109-NOG-Δfrm-C1-PHB-NOG	JM109-NOG-frm, pZQ_mdh2-hps-phi and pBHR70	This study

### 2.6. Gene Expression in E. coli and Protein Purification

The *E. coli* strain BL21 (DE3) with constructed pET-28a vectors grew in lysogeny broth (400 mL; Sangon Biotech Co., Ltd., Shanghai, China) at 37 °C and 200 rpm with 50 µg/mL kanamycin (Solarbio Science & Technology Co., Ltd., Beijing, China) to an absorbance ranging from 0.6 to 0.8 (600 nm). Gene expression was induced by adding 0.1 mM isopropyl-b-D-thiogalactopyranoside (Solarbio Science & Technology Co., Ltd., Beijing, China), and the cells were incubated at 18 °C for 16 h. The bacterial cells were harvested by centrifugation (10,000× *g* for 30 min at 4 °C) and re-suspended in a lysis buffer (20 mM imidazole, 50 mM Tris HCl, 150 mM NaCl (Solarbio, Beijing, China; pH = 7.5)). The cells were disrupted by passing the cell suspension three times through a high-pressure homogenizer (JN-3000 PLUS) at 6,894,757 Pa (1000 Psi). Cell debris and membrane fractions were removed by centrifugation (10,000× *g* for 30 min at 4 °C). The supernatant was loaded onto a Ni–NTA agarose column (GE Healthcare Bio-Sciences AB, Uppsala, Sweden) equilibrated with the lysis buffer (20 mM imidazole, 50 mM Tris HCl, 150 mM NaCl; pH = 7.5). The column was washed with 2 column volumes of the lysis buffer and 2 column volumes of the wash buffer (50 mM imidazole, 50 mM Tris HCl, 150 mM NaCl; pH = 7.5). The proteins were finally eluted with a gradient elution buffer ((100 mM, 150 mM, 200 mM, respectively) imidazole, 50 mM Tris HCl, 150 mM NaCl; pH = 7.5). The purified proteins were desalted using an ultrafiltration tube (10,000 MWCO; Millipore, Burlington, MA, USA) with a desalting buffer (50 mM Tris HCl, 150 mM NaCl, 6 mM MgCl_2_; pH = 7.5) [31]. Protein concentration was determined using the Bradford method [32] with a BCA Protein Assay Kit (Solarbio, Beijing, China), and the sizes of the standard and protein enrichments were confirmed through sodium dodecyl sulfate–polyacrylamide gel electrophoresis (SDS–PAGE).

### 2.7. Enzyme Activity Measurement

#### 2.7.1. Assay of NAD-Dependent Methanol Dehydrogenase Activity In Vitro

The reaction was conducted in 3-paralleled 200 µL in pre-heated (37 °C) 50 mM K_2_HPO_4_ buffer containing 5 mM MgSO_4_, 500 µM NAD^+^ (all purchased from Sigma-Aldrich, St. Louis, MO, USA), and 200 µg protein; 0.5 M methanol (Sigma-Aldrich, St. Louis, MO, USA) was added [10]. The production of NADH was measured at 340 nm and 37 °C with a microplate reader (Infinite 200 PRO; Tecan, Zurich, Switzerland) every minute until a steady state was reached. Average values were used to calculate the activity of enzymes in vitro according to a standard curve of NADH at different concentrations combining the reading sample time. One unit (U) was defined as the amount of enzyme required to process 1 mM of substrate per minute, and specific units (U/mg protein) were calculated by the total protein concentration [10].

#### 2.7.2. Assay of Hps–phi Fusion Activity In Vitro

The fused enzyme activity of purified Hps–phi was assayed by measuring the NADPH formation via coupled reactions catalyzed by glucose-6-phosphate dehydrogenase (G6PD) and phosphoglucoisomerase (Pgi) at 37 °C as described previously [33,34]. Subsequently, 5 mM ribose-5-phosphate (Sigma-Aldrich, St. Louis, MO, USA) was used as the initial substrate to convert the reaction mixture of 200 µL to ribulose-5-phosphate by 10 U Pgi at 37 °C for 5 min to start the reaction. The reaction mixture contained 50 mM K_2_HPO_4_, 2.5 mM NADP^+^, 5 mM MgSO_4_, 10 U G6PD, and 10 U Pgi. To start the reaction, 5 mM formaldehyde was added to the reaction mixture. Each sample was tested in 3 parallels.

#### 2.7.3. Enzyme Activity Measurement by Formaldehyde Production/Consumption In Vivo

Enzyme activity assay for methanol dehydrogenase and Hps–phi was conducted in vivo using a method similar to that described in the study by Muller et al. [10]. Cells were grown overnight in a Murashige and Skoog (MS) medium supplemented with 0.5% ribose (We tested whether ribose could produce more pentose as ribulose could not be purchased. Pentose was preferable for methanol assimilation; data not shown.) and antibiotics. Up to 200 µL supernatant was mixed with Nash’s reagent [35], and the formaldehyde concentration was determined at 412 nm with a microplate reader. The activity was calculated similarly to NAD-dependent methanol dehydrogenase activity in vitro. One unit (U) was defined as 1 mmol formaldehyde produced per minute. For calculating specific units (U/mg protein), the protein concentration was estimated based on the OD_600_ of the culture and the assumption that 1 L culture with an OD_600_ of 1 contains 0.250 g of biomass, half of which was assumed to be protein.

#### 2.7.4. PHB Production and Assay of PHB

Constructed *E. coli* strains were grown overnight in an LB medium with antibiotics. The following morning, 1 mL bacteria solution was used to inoculate in main cultures containing 50 mL of MS medium with 100 mM ribose (Sigma-Aldrich, St. Louis, MO, USA) in 250 mL baffled shake flasks to an OD_600_ of 0.8. Subsequently, 0.1 mM IPTG was added to the solution. Cells were grown at 20 °C, shaking at 220 rpm for 10 h.

Cells were centrifuged at 10,000× *g* for 4 min at room temperature and re-suspended with water. They were then centrifuged and re-suspended twice. The pellet was re-suspended in 50 mL standard MS medium without glucose or ribose, and the actual OD_600_ was measured. The re-suspended cells were inoculated at 37 °C, shaking at 150 rpm as 0.5 M methanol was added. After inoculation for 22 h, the re-suspended cells (pelleted by centrifugation at 10,000× *g* for 4 min at 4 °C) were freeze-dried to determine the PHB concentration.

The PHB content (*w*/*w* %) was quantified by gas chromatography (GC5; Persee, Beijing, China) with an HP-5 capillary column [36,37]. The PHB content (*w*/*w* %) was defined as the percent ratio of PHB concentration (dry weight per liter of culture broth) to cell concentration (dry weight per liter of culture broth).

## 3. Results

### 3.1. Prediction of Optimal PHB Production Pathway

The optimal PHB pathway and flux distribution from RuMP and MCC calculated by the extended *E. coli* metabolic model iML1515 is shown in Figure 1. After adding the RuMP pathway (Mdh, Hps-phi) and PHB synthetic pathway (PhaA, PhaB and PhaC) to iML1515, the PHB production rate was calculated as 1 mmol/gDCW/h PHB (C4) from 6 mmol/gDCW/h methanol (C1) (Figure 1A). Methanol was converted to PHB with carbon loss as CO_2_ at the pyruvate decarboxylation step, leading to the C-mol yield of PHB of 0.67 using the RuMP pathway. After adding the NOG pathway (Fxpk) to form the MCC pathway, the PHB production rate was improved to 1.5 mmol/gDCW/h from 6 mmol/gDCW/h methanol consumption (Figure 1B). This rate was equal to 1 C-mol/C-mol carbon yield, which was a 50% increase relative to a C-mol yield of 0.67 obtained from the RuMP pathway. This result implied that the combination of the NOG pathway with the RuMP pathway could theoretically improve PHB yield from methanol in *E. coli*.

### 3.2. Evaluation of the Effect of frmA Deletion on the Degradation of Formaldehyde

Before the introduction of the C1 assimilation module, the endogenous GSH-dependent formaldehyde dehydrogenase was inactivated by deleting its encoded *frmA* gene [38,39], which could prevent formaldehyde from being dehydrogenized to formate, potentially increasing the amount of formaldehyde entering the MCC pathway. The results shown in Figure 2 indicate that formaldehyde concentration in the wild-type JM109 strain decreased rapidly because of the formation of formate by formaldehyde dehydrogenase. By contrast, formaldehyde concentration in the *E. coli* JM109-Δfrm strain did not change over time. Therefore, the following C1 utilization strains were all based on the *frmA* deletion mutant.

### 3.3. Construction of C1 Assimilation Module

The C1 assimilation module included methanol dehydrogenase and the fused Hps–phi enzyme. The Hps–phi enzyme was chosen based on the study by Orita et al. [24], which reported a considerably high activity. The Hps–phi activities measured in vitro and in vivo (4400 and 1602 mU/mg, respectively; Table 3) in our study were similar to the previously reported values. By contrast, the reported Mdh enzyme activities were very considerably low [10]. To find an Mdh enzyme with high activity, we tested three Mdh enzymes from *B. methanolicus* MGA3, *T. acidophilum* DSM 1728, and *C. necator* N-1, respectively. The results shown in Table 3 indicate that Mdh from *B. methanolicus* (encoded by the *mdh2* gene) exhibits the highest activity both in vitro (11.4 mU/mg) and in vivo (9.2 mU/mg). Therefore, *mdh2* was selected to construct the C1 assimilation module in the engineered strains.

To evaluate the effect of the C1 assimilation module, genes encoding the enzymes Mdh and fused Hps–phi were orderly integrated into an expression vector pZQ to generate vectors pZQ_mdh2 and pZQ_mdh2-hps-phi (Table 1). Then these two vectors were transformed into JM109-Δfrm to construct the JM109-Δfrm-mdh2 and JM109-Δfrm-C1, respectively. Then, formaldehyde concentrations for different strains in methanol media were measured (Figure 3). The results indicated that the introduction of *mdh2* could successfully enable the conversion of methanol to formaldehyde in strain JM109-Δfrm-mdh2, whereas the formaldehyde could not be further utilized. With the further introduction of Hps-phi fusion, the formaldehyde concentration could be reduced in strain JM109-Δfrm-C1, which indicated that formaldehyde was converted into the RuMP pathway. However, a significant amount of formaldehyde remained that could be utilized. This might be owing to the lack of ribulose-5-phosphate, which was a key intermetabolite for the condensation of formaldehyde (Figure 1). As we failed to obtain a relatively inexpensive Ru5P as a carbon source, an alternative method using ribose to supply ribulose-5-phosphate, ribose-5-phosphate, and other circulating C5 metabolites by cellular metabolism was tested. An obvious decrease in formaldehyde concentration was observed in strain JM109-Δfrm-C1 using ribose and methanol as carbon sources. This demonstrated that the expression of enzyme Hps–phi fusion was effective for the conversion of formaldehyde into the central pathways when circulating C5 metabolites were relatively sufficient.

### 3.4. Evaluation of the Effect of C1 and NOG Module on PHB Production from Methanol

The construction of plasmid pBHR68 for PHB production (including *phaA*, *phaB*, and *phaC*) was conducted in a previous study [26]. We used the plasmid pBHR68 as the PHB production module. This plasmid was transformed into JM109-Δfrm and JM109-Δfrm-C1 to obtain new strains JM109-Δfrm-PHB and JM109-Δfrm-C1-PHB, respectively. The strain JM109-Δfrm-PHB, which contained only the PHB production module but not the C1 module, was also created as a control.JM109-Δfrm-C1-PHB contained all the required genes for converting methanol to PHB. As shown in Figure 1, the C1 assimilation module needed to combine well with the NOG module (phosphoketolase) to form the complete carbon-conserved pathway. We then further introduced the phosphoketolase gene *fxpk* from *Bifidobacterium adolescentis* into the genome of JM109-Δfrm-C1-PHB, obtaining the strain JM109-NOG-Δfrm-C1-PHB. To increase the Fxpk expression, the fxpk gene was also inserted into the PHB synthesis plasmid pBHR68 for the formation of a new plasmid, pBHR70. Plasmid PBHR70, replacing PBHR68, was introduced into JM109-NOG-Δfrm-C1 to obtain a new strain JM109-NOG-Δfrm-C1-PHB-NOG with improved phosphoketolase expression.

To evaluate whether the C1 and NOG modules contribute to PHB biosynthesis, PHB production was conducted with or without methanol (Figure 4). A statistical analysis of the obtained data was performed using SPSS 18.0 for Windows, and differences between the production with or without methanol were evaluated using a *t*-test, where *p* < 0.05 was considered significantly different. When the C1 absorption function module was absent, the addition of methanol would not increase the yield of PHB in strain JM109-Δfrm-PHB. By contrast, the content of PHB in strain JM109-Δfrm-C1-PHB increased by 7.4% (from 3.27% to 3.51%) when supplied with methanol. Moreover, the PHB content of JM109-Δfrm-C1-PHB strain on methanol (3.51%) was higher than that of the control group JM109-Δfrm-PHB (3.39%). These results indicated that methanol was indeed converted to PHB with the addition of the C1 module. However, both the increases in PHB content were relatively low, implying that the classic RuMP pathway, solely, might not be sufficient for efficient PHB biosynthesis from methanol in *E. coli*.

After the introduction of the NOG module, the PHB content of JM109-NOG-Δfrm-C1-PHB strain in the methanol group (3.75%) was about 22.5% higher than that in the non-methanol group (3.06%). The PHB content of strains with the NOG module (3.75%) was 6.8% higher than that of strains without the NOG module (3.51%) with methanol as a carbon source, which indicated that the addition of the NOG module could facilitate PHB biosynthesis from methanol. With the further overexpression of the NOG module using multi-copy plasmid in strain JM109-NOG-Δfrm-C1-PHB-NOG, its PHB content from methanol (6.19%) was 85% higher than that without methanol (3.33%) and was 65.1% higher than that produced by JM109-NOG-Δfrm-C1-PHB (3.75%). This indicated that the expression level of NOG is important in the methanol bioconversion using the MCC pathway. The comparison of the above results shows that the overexpression of the MCC pathway can significantly facilitate PHB biosynthesis from methanol, which is consistent with the in silico calculation result.

## 4. Discussion

Considering the low cost of methanol and the continuing efforts to convert abundant natural gas and coal into methanol, developing technologies to convert methanol into other industrial products has drawn interest [12]. In recent decades, several studies have reported on the use of methylotrophs for industrial chemical production from methanol [4,5,6,7].

Muller et al. [10] were the first to demonstrate the possibility of using methanol in engineered *E. coli* by introducing the Mdh and RuMP genes into the host. Whitaker et al. [8] further optimized the engineered strain using a superior Mdh from *B. stearothermophilus* and demonstrated that the engineered strain could convert methanol to flavanone naringenin. However, both studies used the natural RuMP pathway for methanol assimilation, which is not favorable for the production of acetyl-CoA–derived products (such as PHB) resulting from carbon loss during pyruvate decarboxylation, as described by Bogorad et al. [11].

On the basis of previous studies and guided by the optimal pathway for PHB production determined by genome-scale network analysis, we first engineered the MCC pathway in *E. coli* to improve product yield. The MCC pathway is the combination of the RuMP pathway with the NOG pathway [40], which can theoretically enable 100% carbon conservation from methanol to acetyl-CoA–derived products.

To evaluate whether the MCC pathway is better than the RuMP pathway in C1 utilization, we first selected the candidate enzymes by testing the activities of the enzymes (Table 3) in vivo and in vitro. As a result, *mdh2* from *B. methanolicus* MGA3 and Hps–phi fusion [24] were selected as C1 assimilation enzymes in this study. Compared with previously reported results [9,24], the results on the enzyme activities in the present study are similar; the candidate enzymes we selected would be effective for the host. The *frmA* gene coded the enzyme converting formaldehyde to formate would greatly reduce the availability of formaldehyde to Hps–phi (Figure 3) if the gene existed. Three types of modules, C1 assimilation modules, NOG modules, and PHB production modules (Figure 1), were then constructed. The three modules could build the MCC pathway in the host with corresponding enzymes (Table 1), and the effect of the C1 assimilation module was confirmed (Figure 4).

The construction of the MCC pathway has also been conducted in vitro [11], which could continuously produce ethanol. However, it could not achieve the complete conversion to products because of the accumulation of intermediary metabolites. In the present study, the formaldehyde generated from methanol could not be completely consumed further (Figure 3). This might be due to the insufficient supply of circulating metabolites, such as ribulose-5-phosphate and xylulose-5-phosphate, which have been reported to be an important factor in methanol bioconversion via the RuMP pathway [9]. Since the construction of *E. coli* strains that can grow with methanol as the sole carbon source is still difficult [9], in this study, the PHB biosynthesis from methanol was conducted via whole-cell biocatalysis (Figure 4). Although the final PHB content from methanol (6.19%) was not as high as those previously reported (ranging from 20% to 29% PHB content) from sugar-based carbon sources [26] or other carbon sources [41,42,43], a significant increase of 65% in PHB content with the MCC pathway was observed relative to that with only the RuMP pathway through statistical analysis. The experimental results confirm the importance of the combination of NOG with RuMP to form the complete MCC pathway, which can obviously facilitate the biosynthesis of acetyl-CoA–derived chemicals from one-carbon compounds. Future efforts can be expected to engineer cellular metabolism for growable strains via the MCC pathway using a one-carbon compound as the sole carbon source.

## 5. Conclusions

On the basis of metabolic network analysis, we found that the MCC pathway can convert C1 compounds into acetyl-CoA–derived products, such as PHB, without carbon loss. We engineered this pathway in *E. coli* by introducing the heterologous genes for C1 assimilation, phosphoketolase (NOG), and PHB production from different organisms. Compared with the strain containing only the RuMP pathway for C1 assimilation, the strain with the complete MCC pathway produced more PHB by assimilating methanol. Despite the low PHB content relative to that from strains growing on glucose, the results indicate that the introduction of the MCC pathway in *E. coli* can improve its ability for C1 compound bioconversion. To our knowledge, the present study is the first metabolic engineering research on the conversion of methanol to PHB in *E. coli*. This study provides a basis for subsequent studies to further improve product yield from methanol.

## Figures and Tables

**Figure 1 bioengineering-10-00415-f001:**
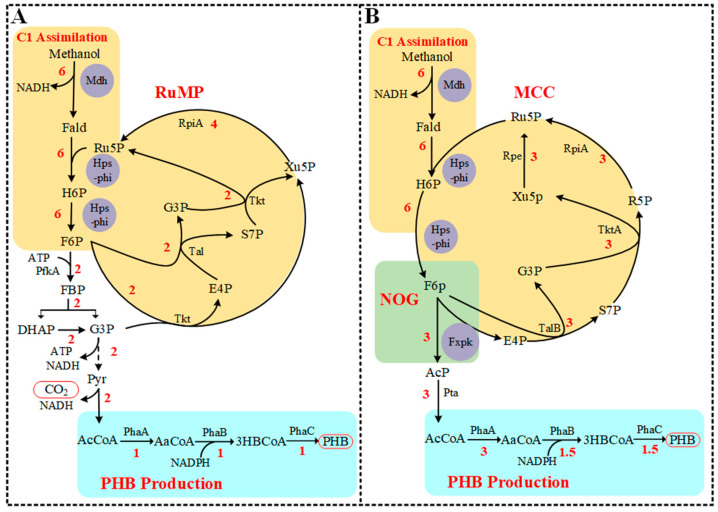
Conversion of methanol to PHB by RuMP pathway (**A**) and MCC pathway (**B**) calculated using flux balance analysis of an extended *E. coli* metabolic network. The C1 assimilation module contained enzymes Mdh (methanol dehydrogenase) and Hps-phi fusion (the fusion protein of hexulose-6-phosphate synthase and 6-phospho-3-hexuloisomerase). The NOG module contained the enzyme Fxpk (phosphoketolase). The PHB production module contained *phaA* (3-ketothiolase), *phaB* (acetoacetyl-CoA reductase), and *phaC* (PHB synthase). Numbers indicate the flux distribution of PHB synthesis from methanol. RpiA, ribose-5-phosphate isomerase A; Tkt, transketolase; Tal, transaldolase; Rpe, ribose-5-phosphate 3-epimerase; TktA, transketolase A; TalB, transaldolase B; Fald, formaldehyde; H6P, hexulose-6-phosphate; F6P, fructose-6-phosphate; FBP, fructose 1,6-bisphosphate; G3P, glyceraldehyde 3-phosphate; DHAP, dihydroxy acetone phosphate; Pyr, Pyruvate; AcCoA, acetyl-CoA; E4P, erythorse 4-phosphate; Ru5P, ribulose-5-phosphate; R5P, ribose-5-phosphate; Xu5P, xylulose-5-phosphate; sedoheptulose-7-phosphate; AaCoA, Acetoacetyl-CoA; 3HBCoA, 3-Hydroxybutanoyl-CoA.

**Figure 2 bioengineering-10-00415-f002:**
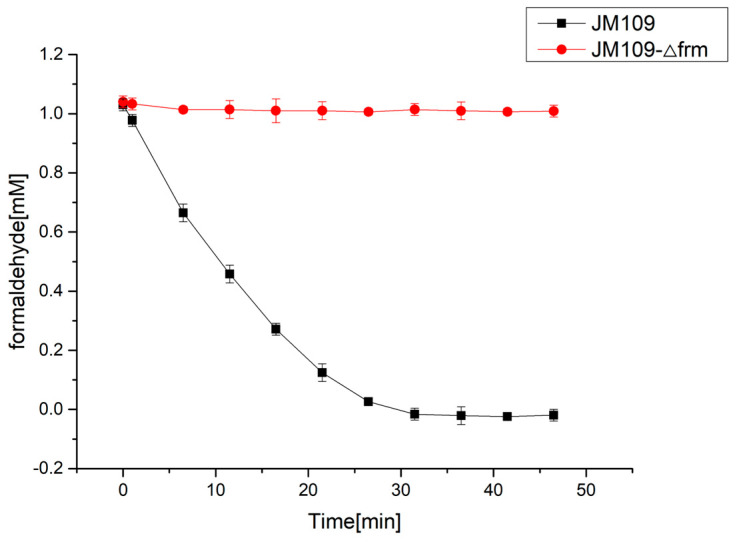
Formaldehyde degradation in *E. coli* JM109 strain (black symbols) and JM109-Δfrm strain (red symbols). The result indicated that the deletion of *frmA* is important to accumulate enough formaldehyde as the substrate for RuMP pathway. All experiments were conducted in triplicate. The mean values of three biological replicates are presented, and error bars indicate standard deviations.

**Figure 3 bioengineering-10-00415-f003:**
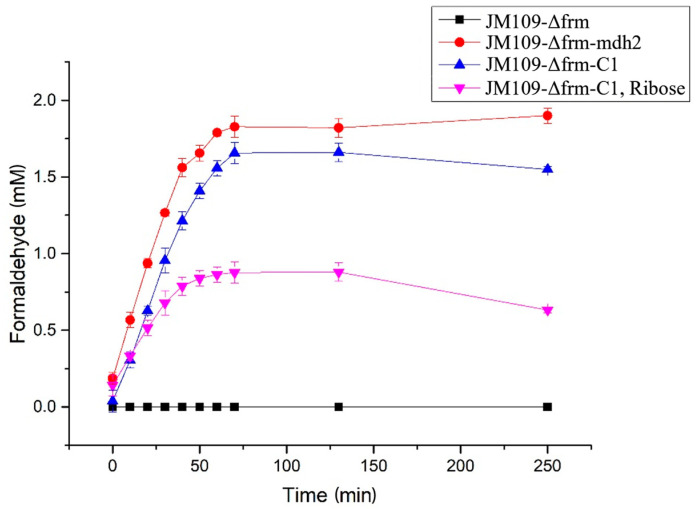
Formaldehyde production in JM109-Δfrm cells expressing *mdh2* in the presence or absence of *hps-phi* fusion. The strain JM109-Δfrm (black squares), which did not contain constructed plasmids, was set as control. The strain JM109-Δfrm-mdh2 (red circles), which contained Mdh2 enzyme, was set to test whether Mdh2 could convert methanol to formaldehyde. The strain JM109-Δfrm-C1 (blue triangles), which contained enzymes Mdh2 and Phs–phi fusion, was set to test whether formaldehyde generated from methanol could be further assimilated into RuMP pathway. Ribose was added as an alternative method to supply circulating C5 metabolites (pink inverted triangles). The mean values of three biological replicates are shown, and error bars indicate standard deviations.

**Figure 4 bioengineering-10-00415-f004:**
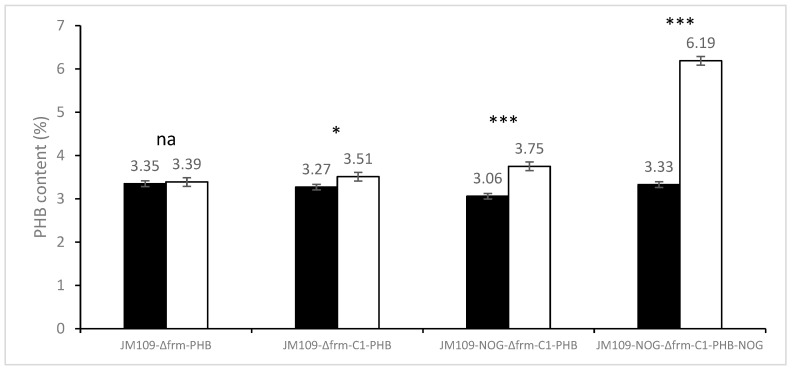
PHB production from different engineered *E. coli* strains. The introduction of the phosphoketolase gene to form the complete MCC pathway greatly improved PHB content. JM109-Δfrm-PHB with only PHB production module was set as control. JM109-Δfrm-C1-PHB with PHB production module and C1 assimilation module was set to test the RuMP pathway. JM109-NOG-Δfrm-C1-PHB with PHB production module, C1 assimilation module, and low-expression NOG module was set to test the MCC pathway. JM109-NOG-Δfrm-C1-PHB-NOG with PHB production module, C1 assimilation module, and high-expression NOG module was also set to test the MCC pathway. Shown are the mean values of three biological replicates, and error bars indicate standard deviations. The dark and white columns represented engineered strains conducted using water and methanol, respectively. *t*-tests were conducted to evaluate statistical significance at *p* < 0.05 (*),and *p* < 0.001 (***).

**Table 3 bioengineering-10-00415-t003:** Measured activities of different enzymes from different sources. In vitro activities were measured using purified proteins; in vivo activities were measured using *E. coli* cell suspensions harboring the corresponding enzymes. Description contains the advantages of each enzyme mentioned in the literature. The hyphen (-) indicates activity not detected. Sources column specifies the species source of these enzymes. Shown are the mean values of three biological replicates.

Enzyme Activity (mU/mg)	In Vitro	In Vivo	Description	Sources	Reference
mdh2	11.4	9.2	Higher activity in vivo	*B. methanolicus* MGA3	[10]
TaADH (Ta1316 ADH)	0.9	-	High relative activity in vitro at high temperature	*T. acidophilum* DSM 1728	[23]
mdh-CT	4.8	0.43	Higher activity in vitro	*C. necator* N-1	[22]
Hps-phi	4400	1602	Higher efficiency than a simple mixture of individual enzymes	*M. gastri* MB19	[24]

## Data Availability

Not applicable.

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
