# Peer review of "Engineering Escherichia coli for Poly-β-hydroxybutyrate Production from Methanol"

_bioengineering, 2023, doi:10.3390/bioengineering10040415_

Round 1

Reviewer 1 Report

1. In introduction section, the last lines should represent objectives of the study but authors mentioned the results of this study.

2. Table 2, Table 3 caption should be improved as there are some duplication of sentences.

3. Statistical analysis is missing.

4. Result section should be presented in Past tense, but authors are presenting data in present tense.

5. Update study with latest references on PHB production, some examples are. https://doi.org/10.1007/s13399-022-02762-0

6. References list should be formatted as per journal instructions.

Author Response

  1. In introduction section, the last lines should represent objectives of the study but authors mentioned the results of this study.

Reply: Thanks for suggesting. We have revised the last lines of the introduction section and deleted the results of this study (Lines 65-72).

  1. Table 2, Table 3 caption should be improved as there are some duplication of sentences.

Reply: Thanks for suggesting. We have revised the caption of the tables.

  1. Statistical analysis is missing.

Reply: Thanks for reminding. We have added statistical analysis in Lines 332-335, 364-367 and Fig. 4.

  1. Result section should be presented in Past tense, but authors are presenting data in present tense.

Reply: Thanks for reminding. We have changed tense in result section.

  1. Update study with latest references on PHB production, some examples are. https://doi.org/10.1007/s13399-022-02762-0

Reply: Thanks for reminding. We add some latest references on PHB production in Line lines 65-70 and line 408.

  1. References list should be formatted as per journal instructions.

Reply: Thanks for reminding. We have formatted the references list.

Reviewer 2 Report

The authors present a manuscript on engineering the synthesis of PHB in E. coli using methanol as substrate. Based on models they design and construct a pathway with high accumulation of intermediates and good performance in methanol utilisation and PHB production.

The data shows that the computer model can be used in good reliance to predict effect of deletions and overexpressions of genes. 

They show that PHB can be synthesized at a high level when both genes and precursor formation are optimized. This is key to produce high levels of product. 

Some suggestions and comments:

Table 4: names of species should be added to the table (not just in the text).

Fig. 4: the legend does not explain the columns and the colour of columns. Please add.

Author Response

Table 4: names of species should be added to the table (not just in the text).

Reply: Thanks for suggesting. We have added the species in Table 4.

Fig. 4: the legend does not explain the columns and the colour of columns. Please add.

Reply: Thanks for reminding. We have added the legend explaining of the columns in Fig4.

Reviewer 3 Report

The manuscript by Chen Z. et. al., “Engineering Escherichia coli for poly-β-hydroxybutyrate production from methanol” reported that the introduction of the NOG pathway is essential for improving PHB production (a 65% increase in PHB concentration). However, the studies are fine and updated. There are few concerns to be addressed as follows:

Comments

1.     Lines 32-33, Authors should elaborate on the alternative sources of green biofuels, such as biogas, bioethanol, etc. e.g.  doi.org/10.1002/biot.201800468, doi.org/10.1016/j.biortech.2020.124550

2.     Lines 66-69, The use of PHB is limited due to its fragile nature and is not suitable for biotechnological applications as compared to co-polymers of PHB, i.e., PHA. Please justify the applications.

3.     The authors should clearly state the novelty and significance of this manuscript.

4.     The discussion should be more polished with qualitative data and PHB production to justify the yield productivity.

5.     A perspective on the use of PHB products and their importance in biotechnology applications should be included.

Author Response

  1. Lines 32-33, Authors should elaborate on the alternative sources of green biofuels, such as biogas, bioethanol, etc. e.g.  doi.org/10.1002/biot.201800468, doi.org/10.1016/j.biortech.2020.124550

Reply: Thanks for suggesting. We have added the alternative sources in Lines 33,34.

  1. Lines 66-69, The use of PHB is limited due to its fragile nature and is not suitable for biotechnological applications as compared to co-polymers of PHB, i.e., PHA. Please justify the applications.

Reply: Thanks for suggesting. PHA is true suitable for biotechnological applications. But recent demonstrated that PHB is important not only as a carbon stock but in supporting bacterial survival under adverse conditions (, https://doi.org/10.1093/femsre/fuaa058 ). As the intermediate product, formaldehyde, is toxic to the strains when the accumulation of the product. We think PHB might be helpful for the survival of the strains. In addition, PHB also has very good biocompatibility which is mainly used in drug delivery carriers and biological tissue engineering. We have add its applications in lines 65-70.

  1. The authors should clearly state the novelty and significance of this manuscript.

Reply: Thanks for sugesting. We have added the novelty and significance of this manuscript in conclusion section Lines 425-426.

  1. The discussion should be more polished with qualitative data and PHB production to justify the yield productivity.

Reply: Thanks for suggesting. We have polished in discussion section in Lines 409-410. And some statistical analysis of PHB production were also added in results section (Lines 332-335, Fig. 4).

  1. A perspective on the use of PHB products and their importance in biotechnology applications should be included.

Reply: Thanks for sugesting. We have added some perspective on the use of PHB products in Lines 65-70. In this study, we mainly focus on the feasibility of MCC pathway and the possibility of conversion of methanol in E. coli, and PHB is one the production of acetyl-CoA derived product.